Embedding the value of coastal ecosystem services into climate change adaptation planning

Wedding Lisa M. lisa.wedding@ouce.ox.ac.uk 1 2
Reiter Sarah 2 3
Moritsch Monica 2 4
Hartge Eric 2
Reiblich Jesse 2 5
Gourlie Don 2 6
Guerry Anne 7
1 School of Geography and the Environment, University of Oxford , Oxford , United Kingdom
2 Center for Ocean Solutions, Stanford University , Stanford , CA , United States of America
3 Anderson Cabot Center for Ocean Life, New England Aquarium , Boston , United States of America
4 Department of Ecology and Evolutionary Biology, University of California, Santa Cruz , Santa Cruz , CA , United States of America
5 Virginia Coastal Policy Center, William & Mary Law School , Williamsburg , VA , United States of America
6 Puget Sound Partnership , Seattle , WA , United States of America
7 Natural Capital Project , Stanford , CA , United States of America
Pochon Xavier
Electronic publication date: 2022 Aug 23
Publication date: 2022
Volume: 10
Electronic Location ID: e13463
Received 2020 Apr 2; Accepted 2022 Apr 28
Copyright: ©2022 Wedding et al.
Copyright year: 2022
Copyright holder: Wedding et al.
License: This is an open access article distributed under the terms of the Creative Commons Attribution License, which permits unrestricted use, distribution, reproduction and adaptation in any medium and for any purpose provided that it is properly attributed. For attribution, the original author(s), title, publication source (PeerJ) and either DOI or URL of the article must be cited.
License URL: https://creativecommons.org/licenses/by/4.0/

Keywords: Ecosystem services, Climate adaptation, Coastal habitats, Coastal protection

Funding: The Packard Foundation This work was supported by the Packard Foundation. The funders had no role in study design, data collection and analysis, decision to publish, or preparation of the manuscript.

==============================
Coastal habitats, such as salt marshes and dune systems, can protect communities from hazards by reducing coastline exposure. However, these critical habitats and their diverse ecosystem services are threatened by coastal development and the impacts from a changing climate. Ever increasing pressure on coastal habitats calls for coastal climate adaptation efforts that mitigate or adapt to these pressures in ways that maintain the integrity of coastal landscapes. An important challenge for decisionmakers is determining the best mitigation and adaptation strategies that not only protect human lives and property, but also safeguard the ability of coastal habitats to provide a broad suite of benefits. Here, we present a potential pathway for local-scale climate change adaptation planning through the identification and mapping of natural habitats that provide the greatest benefits to coastal communities. The methodology coupled a coastal vulnerability model with a climate adaptation policy assessment in an effort to identify priority locations for nature-based solutions that reduce vulnerability of critical assets using feasible land-use policy methods. Our results demonstrate the critical role of natural habitats in providing the ecosystem service of coastal protection in California. We found that specific dune habitats play a key role in reducing erosion and inundation of the coastline and that several wetland areas help to absorb energy from storms and provide a protective service for the coast of Marin county, California, USA. Climate change and adaptation planning are globally relevant issues in which the scalability and transferability of solutions must be considered. This work outlines an iterative approach for climate adaptation planning at a local-scale, with opportunity to consider the scalability of an iterative science-policy engagement approach to regional, national, and international levels.

Introduction

Ecosystem services are the stream of vital benefits flowing from natural capital to people (Barbier et al., 2013; Costanza et al., 1997). Coastal habitats—such as seagrass, kelp forests, salt marshes, and dunes—provide benefits that are extremely valuable to society, such as carbon sequestration, nutrient cycling, sustaining biodiversity, tourism and recreation (Agardy, 1993; Barbier et al., 2013; Beck et al., 2018; Beck et al., 2001; Duarte, 2017; Guerry et al., 2012; Pendleton et al., 2011). In addition, coastal ecosystems are also valued for their non-material spiritual, bequest value, emotional, aesthetic, and health benefits (Duraiappah et al., 2005; Ghermandi et al., 2009; Sandifer & Sutton-Grier, 2014).

Coastal habitats also plays a critical role in coastal protection, which directly benefits coastal communities by reducing the effects of coastal flooding and erosion caused by storms and rising seas (Arkema et al., 2015; Barbier et al., 2013; Möller et al., 2014; Narayan et al., 2017; Spalding et al., 2014). Yet, these critical coastal habitats are threatened by existing coastal infrastructure and impacts from a changing climate (Defeo & McLachlan, 2005; Dugan et al., 2011; Guannel et al., 2015; Heady et al., 2018). As coastal development and rising sea levels damage or destroy natural habitats, communities and infrastructure become increasingly vulnerable to storms and erosion (Guannel et al., 2015; Neumann et al., 2015; Nicholls, Hoozemans & Marchand, 1999).

In coastal California, ever increasing pressure on coastal habitats calls for land management and adaptation efforts that mitigate or adapt to these pressures in ways that maintain the integrity of coastal landscapes (California Coastal Commission, 2015). Without coastal climate adaptation efforts that incorporate conservation or restoration of coastal habitats, these ecosystems will continue to be lost, and their protective benefits (together with diverse co-benefits) will disappear with them (Neumann et al., 2015). Maintaining natural capital to protect and support vibrant coastal communities is especially critical in the face of intensifying climate change effects. This effort is no small task, and it presents coastal communities with a significant challenge—and opportunity—to proactively manage land use via protection and restoration of coastal habitats (Caldwell & Segal, 2007; Sutton-Grier et al., 2018).

To foster coastal adaptation, some planners and decisionmakers are considering incorporating a suite of natural or nature-based infrastructure strategies. Nature-based solutions (NbS) work with nature to address an environmental or societal challenge, benefiting humans and biodiversity (Seddon et al., 2021) and will be crucial in addressing the challenges related to climate change adaptation and mitigation, biodiversity loss, and human wellbeing (Seddon et al., 2020). NbS are built systems that combine natural ecosystems with engineered structures to provide added protection as well as multiple other services to communities (Sutton-Grier et al., 2018). Nature-based infrastructure strategies are key components of overall NbS efforts. For example, stream-design culverts can help to reduce damage to property and roads from coastal flooding while restoring natural tidal flow (Gillespie et al., 2014). As nature-based infrastructure strategies gain traction, there is a need to accurately identify suitable locations and appropriate settings for these strategies to ensure long-term delivery of the protective service and additional co-benefits (Temmerman et al., 2013; Arkema & Ruckelshaus, 2017; Ruckelshaus et al., 2016). While hardened shoreline structures can protect infrastructure immediately behind them, the structures can also alter sediment transport regimes, eventually leading to beach erosion in front of and adjacent to armoring (Griggs, 2005; Kraus, 1988).

Adaptation to climate change impacts has gained prominence in scientific and policy agendas (Moser & Ekstrom, 2010) and many governmental and non-governmental actors at the national, regional, and local levels are developing climate adaptation plans. Overcoming climate adaptation barriers involves incremental policy, planning, and management choices (Ekstrom & Moser, 2014; Melius & Caldwell, 2015). California features a relatively prominent policy framework for protecting the state’s shoreline and coastal managers have bolstered this foundation with additional guidance and funding (California Coastal Commission, 2018b; California Coastal Commission, 2015). Adapting to the threats that climate change poses to California’s coastal communities can be addressed through the state’s land use policies. The California Coastal Act (California Public Resources Code, 1976) serves as the state’s coastal management program and legal framework. It was enacted in 1976 to regulate land use and development in the coastal zone—i.e., an area extending seaward three miles and landward according to legally defined boundaries (California Public Resources Code, 1976). The Coastal Act requires local governments in the coastal zone to prepare Local Coastal Programs (LCPs), including land use plans and implementing measures, such as zoning ordinances (California Public Resources Code §§30500-30526). The California Coastal Commission reviews and approves LCPs as consistent with—and adequate to carry out—Coastal Act policies, after which that local government becomes the lead agency for permitting most coastal development above the mean high tide line, subject to limited California Coastal Commission appeal authority (California Public Resources Code §§30514a-30514b). Thus, LCPs are a critical decision and entry point for local-level coastal climate adaptation actions (Caldwell & Segal, 2007). LCP updates are one substantial policy mechanism for local governments to address coastal climate adaptation in California (Berke & Lyles, 2013).

Here, we set out to advance the understanding of how natural habitats reduce vulnerability of coastal assets (e.g., infrastructure, parks, habitats) and analyzed the legal and policy considerations relevant to the California Local Coastal Programs update process. Further, we incorporated the California Coastal Commission’s sea-level rise policy recommendations in our coastal vulnerability modeling efforts to assist Marin county in developing approaches that integrated current ecosystem service science where suitable. Here, we utilized an ecosystem-service modeling approach to ask the following: (1) What is the role of natural habitat in providing the ecosystem service of coastal protection? and (2) Where are coastal habitat locations that might be prioritized for restoration and management in order to reduce risk to coastal ecosystems, people and property? We determined the role of natural habitat in reducing exposure to erosion and inundation throughout the Pacific coast of Marin county, California, USA using the Coastal Vulnerability Model. We also evaluated the extent of these coastal protection benefits by mapping where the resulting estimates of high hazard exposure aligned with various land use zoning designations and identified areas where large numbers of people and property were exposed to coastal hazards. In collaboration with local decision makers, we linked our ecosystem service mapping and assessment to coastal adaptation decision making, and synthesized potential nature-based strategies relevant to these local coastal communities.

Materials & Methods

Study area

The Pacific coast of Marin county, California includes extensive natural habitats that provide a suite of ecosystem services (Fig. 1). Through direct dialogue with members of the County of Marin Community Development Agency, we identified and then examined two case study areas of particular economic and ecological significance for the county’s coastline: Dillon Beach and the Stinson Beach-Bolinas Lagoon area. Dillon Beach is in Marin’s northernmost coastal community with a suite of habitats including predominantly dune systems and surf grass. These natural habitats influence the cultural attachment to the coast for Dillon Beach residents and visitors alike (Tierney, 2017). For example, the coastal areas provide recreation through beach use, camping, bird watching, fishing, boating, and surfing (Barbier et al., 2013; Tierney, 2017). Further, the coastal ecosystems in this area provide critical habitat for a seabird colony and support two marine mammals haul out locations that provide rest between foraging (Hayden et al., 2017). Natural habitats of Stinson Beach and Bolinas Lagoon include primarily coastal surf grass and kelp habitat, with wetland habitat in the tidal embayment of Bolinas Lagoon, and a low dune system along Stinson Beach sandspit. Bolinas Lagoon shelters a predominantly saline, shallow water mosaic of mudflats, riparian areas, and tidal salt marsh that covers approximately 4.5 km2. Wetlands in Bolinas Lagoon help coastal areas absorb energy from storms and provide a protective service for the adjacent lagoon shoreline. Bolinas Lagoon is a “Wetland of International Importance” (Ramsar Convention, 2018) and provides critical habitat for wintering shorebirds along the Pacific Flyway.

Figure 1 Coastal habitats of Marin county that can confer protection from coastal hazards such as inundation and erosion.

Habitats include kelp, wetlands, eelgrass, surfgrass, and sand dunes. Grey lines denote county boundaries.

Ecosystem service mapping and assessment

We used the InVEST (Integrated Valuation of Ecosystem Services and Tradeoffs) Coastal Vulnerability model (Sharp et al., 2018) to evaluate the role that coastal habitats play in reducing exposure to erosion and flooding by comparing the exposure index value of a given coastal segment with habitats present and with habitats absent. The model can account for both service supply (e.g., natural habitats as buffers for storm waves) and the location and activities of people who benefit from services (e.g., the location of people and infrastructure potentially affected by coastal storms). The InVEST Coastal Vulnerability model produced a numeric Exposure Index (EI), which ranges from 1 to 5 (5 = highest risk; 1 = lowest risk). This index provided a ranked estimate of which coastline segments demonstrated relatively high or low exposure to coastal erosion and inundation due to sea-level rise and storms. While this index is relative and does not calculate absolute probabilities of erosion and inundation, it provides a heuristic way of comparing coastal segments and highlights areas where multiple conditions creating high exposure to hazards coincide. In particular, this model can illustrate the effects of relative differences in protection conferred by hardened shoreline structures versus natural and nature-based alternatives through a coastal exposure index. Further, the model can examine the relative impact of conserving, restoring, or destroying different habitat types at any given location.

The Coastal Vulnerability model data inputs served as proxies for various complex shoreline processes that influence exposure to erosion and inundation. The data inputs included: a polyline with attributes about local coastal geomorphology along the shoreline, polygons representing the location of natural habitats (e.g., seagrass, kelp, wetlands, etc.), rates of projected net sea-level change, a depth contour that could be used as an indicator for surge level (edge of the continental shelf), a digital elevation model that represented the topography and bathymetry of the coastal area, and a point shapefile that contained values of observed storm wind speed and wave power (Table 1). The protective capacity of natural habitats within a specified distance of the coastline (Table S2) confered protection from coastal hazards to adjacent areas. When multiple habitats were present, this protection increased nonlinearly and caused input risk ranking to decrease (Supplemental Information). While the model does not account for possible changes in the coastline shape over time, the EI incorporates which habitats are most likely to experience coastal erosion and flooding. The model provided a relative estimate of exposure under different land use scenarios (Supplemental Information).

Table 1 The Coastal Vulnerability model data inputs serve as proxies for various complex shoreline processes that influence exposure to erosion and inundation.

	Data input	Data description	Data source	
Geomorphology	A line shapefile input was used to calculate the Geomorphology ranking of each section of shoreline	A polyline with attributes about local coastal geomorphology along the shoreline	NOAA Environmental Sensitivity Index	
Coastal habitat	The model used the input layers to calculate a Natural Habitat ranking for each shoreline segment	Polygons representing the location of coastal habitats	California Department of Fish and Wildlife website created for Marine Life Protection Act	
Wind and wave exposure	Wind and wave data were given in a grid of points spaced approximately 50 km apart off the coast of Marin county	A point shapefile containing values of observed storm wind speed and wave power across areas of interest. For each point, the risk ranking was based on the top 10% of values for wind speed and wave height	WaveWatch III data, provided by NOAA	
Surge potential	Distance from shore to the edge of the continental shelf was used as a proxy for oceanic surge distance; longer distances between the coastline and edge of shelf results in higher storm surges	A polyline of the edge of the continental shelf around the North American west coast	InVEST Coastal Vulnerability Model data download materials (Sharp et al., 2018)	
Relief	An elevation raster was used to determine low-lying coastal areas	5 m resolution bathymetry/topography digital elevation model of California’s coastal land and waters	United States Geological Service (Foxgrover & Barnard, 2012)	
Sea level rise	The upper range of SLR projections were used as a precautionary approach	Rates of projected net sea level change up to 2030 were informed from local variation in global SLR and coastal land subsidence/uplift rates	National Research Council (2012)	

We used data specific to California for geomorphology, coastal habitat area, rate of sea-level rise through 2030 (Table 1). We coupled these data with global models of wind and wave power (Wave Watch III), and the edge of the continental shelf (surge potential). The geomorphology data input was represented by a polyline with attributes about local coastal geomorphology along the shoreline based on the NOAA Environmental Sensitivity Index (Peterson, 2002). In order to account for locations of armoring from man-made structures, we used an inventory of barriers that have potential to retain sandy beach area from the California Coastal Commission (2014). Due to changes in hydrodynamics, soft sediment areas adjacent to hardened barrier structures are highly likely to erode (Kraus, 1988), so we altered the geomorphology rank of coastal segments that were within 75 m of (but not directly behind) armoring structures to reflect this increased risk (Table S1). Polygons that represented the location of natural habitats (e.g., seagrass, kelp, wetlands, etc.) were obtained from the California Department of Fish and Wildlife (California Department of Fish and Wildlife, 2016; https://data-cdfw.opendata.arcgis.com/). A point shapefile that contained values of observed storm wind speed and wave power across an area of interest was created using Wave Watch III data provided by NOAA. A polyline of the edge of the continental shelf serves as a proxy for oceanic surge potential. In general, a longer distance between the coastline and the edge of the continental shelf will result in a higher storm surge. The model does not account for land barriers in front of coastal segments that would alter storm surge. A 5-meter resolution bathymetry/topography digital elevation model of California’s coastal land and waters from the United States Geological Service (Foxgrover & Barnard, 2012) was used with mean sea-level datum at 0-m. The rate of projected net sea-level change through 2030 was derived from local variation in global sea-level rise and coastal land subsidence/uplift rates (National Research Council, 2012).

We mapped the benefits of natural habitats in reducing exposure to coastal impacts throughout the Pacific coast of Marin county. We also evaluated these benefits in key areas of local importance, including Dillon Beach and the Stinson Beach-Bolinas Lagoon area. The model produced a qualitative estimate of risk in terms of an EI for every 250 m segment of coastline. The EI differentiates areas with relatively high or low exposure to coastal erosion and inundation during storms. By coupling these results with coastal features of interest (e.g., infrastructure, land use zoning, or population), the model identified areas along a given coastline where people or property are most vulnerable to storm waves and surge. EI values were assigned classifications of “High” exposure, “Medium” exposure, and “Low” exposure based on percentile ranks in the overall EI distribution (Table S1). We classified the role of habitats in reducing EI values using the same percentile ranks (Supplemental Information). We mapped where the resulting estimates of high hazard exposure aligned with various land use zoning designations in Marin county (Marin County Community Development Agency, 2015) and identified areas where large numbers of people and property were exposed to coastal hazards. All modeling was performed with InVEST version 3.3 (Sharp et al., 2018), and all other geospatial operations were performed with ArcMap 10.1 (ESRI).

Policy context and analysis

Modeling and mapping of ecosystem services can support the assessment of place-based coastal protection services provided by coastal habitats and support science-based climate adaptation strategies (Arkema et al., 2017; Arkema et al., 2013). Throughout the duration of the ecosystem service mapping, the research team conducted iterative engagement discussions with members of relevant coastal planning agencies, local communities, as well as trusted external collaborators (Fig. 2). For example, we were able to provide mapped visual products and initial synthesized results to Marin county planners as they conducted a series of community engagement meetings to enable iterative engagement and feedback from stakeholders. In one instance, we directly participated in a meeting with community members to provide additional context regarding the ecosystem services approach and engaged in a discussion about the potential implications from the results. In addition, members of the research team participated in regional dialogues, such as in projects for the Farallones National Marine Sanctuary (Hutto, 2016), to gain a deeper understanding on coastal adaptation topics relevant to the collaborators in Marin county. These direct engagement opportunities provided our policy research team with additional considerations of interest from local community members, which improved the applicability of the adaptation policy findings. Further, this approach allowed for the ecosystem service mapping and modeling to be presented and discussed with a range of stakeholders during the project. Through these series of interactions, the collective research team was able to refine some of the coastal vulnerability modeling data inputs and assumptions based on external feedback and thus ensure a higher likelihood of uptake for the resulting findings.

Figure 2 Overview of iterative stakeholder engagement during the ecosystem service mapping approach.

Through these direct engagement opportunities, the collective research team was able to refine the analytical approach based on external feedback and thus ensure a higher likelihood of uptake for the resulting findings.

To connect the science to policy, academic literature and practitioner guidance was evaluated to identify potentially appropriate coastal adaptation strategies for sea-level rise. We reviewed guidance documents and reports that outline land use planning and regulatory options that could be considered in coastal areas (Grannis, 2011; Siders, 2013). We also researched relevant state- and county-level laws and policies on acceptable strategies for near- and long-term adaptation to coastal hazards. The identification of relevant laws and policies stemmed from iterative engagement with agency staff at the state and county levels as well as with legal experts familiar with the topics at the state and local levels. We identified the legal and practical limitations these policies place on adaptation options in Marin and explored potential changes to the existing policies that may increase adaptive capacity. In each of the case study locations, we identified near-term natural or nature-based coastal adaptation strategies that could maintain or enhance existing coastal protection services. These comparisons were informative when evaluating the coastal protection benefits and tradeoffs among adaptation strategies. Specifically, we assessed exposure to coastal hazards and adaptation options of case study locations in the county to better understand the different dimensions of these vulnerabilities.

Results

Coastal exposure and the role of natural habitat

Across the Marin coastline, areas of wetlands and dune habitat provided varying degrees of coastal protection from storms and sea-level rise. Overall, these natural habitats provided the highest degrees of protection from coastal hazards along the northern shore of Point Reyes and around Dillon Beach (Fig. 3B). Specifically, the high dune habitat at Dillon Beach was found to aid in protecting important roads and the small community at Lawson’s Landing, while also providing key recreational beach going and camping opportunities. The surf grass along the agricultural areas bordering Estero de San Antonio were found to provide a lower relative role in reducing exposure to coastal impacts. In addition, the low dune system along Stinson Beach and near the mouth of Bolinas Lagoon played a medium role in reducing exposure to erosion and inundation from storms compared to the rest of the Marin coastline (Fig. 4B).

Figure 3 (A) Coastal habitats around Dillon Beach that confer protection from coastal hazards such as inundation and erosion. (B) The relative role of coastal habitats around Dillon Beach in reducing exposure to erosion and inundation from storms (darker colors denote a greater role).

Relevant land use zoning information is included. Specifically, dunes aid in protecting important roads and the small community at Lawson’s Landing while also providing key recreational beach going and camping opportunities. The surfgrass along the agricultural areas bordering Estero de San Antonio play a lower relative role in reducing exposure to coastal impacts. Role of habitats is relative to the entire coast of Marin county.

Figure 4 (A) Coastal habitats around Bolinas, Bolinas Lagoon, and Stinson Beach that confer protection from coastal hazards. (B) The relative role of coastal habitats around Bolinas and Stinson Beach in reducing exposure to erosion and inundation from storms (darker colors denote a greater role).

Relevant land use zoning information is included. Specifically, the mouth of Bolinas Lagoon and the neighborhood behind Stinson Beach receive the greatest relative protection from coastal beach and dune systems. Role of habitats is relative to the entire coast of Marin county.

The low and high dune systems in the northern portion of Marin county served the highest relative role in reducing exposure to erosion and inundation from storms. Coastal habitats in the southern portion of Marin county provided the lowest protective role (Fig. 5). We used Marin county’s zoning layers (Marin County Community Development Agency, 2015) coupled with the outputs of the InVEST Coastal Vulnerability model to identify how priority or high-exposure locations align with the county’s various land-use or zoning designations. These overlay results informed the type of coastal adaptation strategies most feasible in each location. For example, when high-exposure areas corresponded with residential zoning designations with existing structures they reduced the feasibility of habitat restoration or retreat options that might conflict with private property rights or result in politically challenging debates (Melius & Caldwell, 2015).

Figure 5 The relative role of coastal habitats in Marin county in reducing exposure to erosion and inundation from storms (darker colors denote a greater role).

Relevant State and National Park Lands are included.

The high dune habitat at Dillon Beach (Fig. 3A) served a relatively high role countywide in reducing erosion and inundation of the coastline (Fig. 3B). This area of the county has less than 100-m of hardened structures along the coastline, increasing reliance on natural habitats for protective services. Dune habitats directly in front of the main residential commercial center near Lawson’s Landing reduced the coastal exposure for this area. As the dunes transition to surf grass, we found that the relative coastal protection reduces to intermediate levels. On the opposite side of Tomales Bay, the shoreline benefits from an EI reduction, the largest reduction in the Dillon Beach area (Fig. 4B). Though multiple different habitat types are located in Tomales Bay, only eelgrass was within appropriate proximity of these segments of coastline to confer protection.

Coastal policy options

At Dillon Beach, in the short to medium term, a large-scale dune restoration project is possible on the south end of the beach—near the mouth of Tomales Bay. Here, experimental design areas and monitoring could aid in testing the protective services dunes provide. Dune restoration may help to protect exposed “Residential” parcels (including residential structures) as well as the “Resort and Commercial Recreation” areas and important inland wetland habitat. Marin would be at the forefront of helping to develop data to determine dune restoration design metrics, and elements of success and identifying how hydrological and geomorphological conditions in different areas contribute to the success or failure of restored dunes as a natural infrastructure alternative to armoring. Coastal dune restoration on the west coast of North America was pioneered in the Lanphere Dunes in Humboldt county in the 1980s (Pickart, 2013). A dune restoration project in Dillon Beach could add to the body of evidence from similar demonstration sites previously approved by the Coastal Conservancy.

In Stinson Beach, a primary short-term option is to “hold the line” or protect existing natural and built infrastructure in place by using physical barriers to the sea and applying a hybrid concept in this area. This could include a horizontal levee along Bolinas Lagoon and beach nourishment and/or dune restoration along the Stinson Beach coastline. The horizontal levee could provide significant protection to the western section of Bolinas Lagoon zoned as “Agriculture Residential Planned.” A longer-term option in Stinson Beach is to “adjust to the line” or accommodate the infrastructure by using development conditions and/or restrictions that provide incentives to reduce the exposure of existing or rebuilt infrastructure to increased inundation from storm events. To the extent that other natural habitats in the lagoon can be protected, restored, or enhanced, there could be benefits provided by a horizontal levee project. Zoning designations in the Stinson Beach and Bolinas Lagoon areas limit the availability of policy options. This is because “at risk” areas correspond with a patchwork of high- and low-density housing designations in the Stinson Beach area, generally. However, the western side of Bolinas Lagoon is zoned as open space and residential agriculture planned, thus, the most feasible locations for wetland restoration occur along the western side of Bolinas Lagoon.

Discussion

An important challenge for decisionmakers is determining the best mitigation and adaptation strategies that not only protect human lives and property, but also protect the ability of coastal habitats to provide the broad suite of benefits we rely on (Aerts et al., 2014; Heady et al., 2018). We determined the role of natural habitat in reducing exposure to erosion and inundation throughout the Pacific coast of Marin county, California, USA using a coastal vulnerability model. The InVEST Coastal Vulnerability model allowed us to identify relative exposure to inundation and erosion for coastal settings and identify locations where coastal habitats play a significant role in reducing that exposure (Arkema et al., 2013; Ruckelshaus et al., 2016). Previous studies have reported that coastal habitats (seagrass, mangrove and coral reefs) may have a greater collective effect in reducing coastal vulnerability when they exist near each other, than individual habitats do (Guannel et al., 2016). Nonetheless, individual habitat types are still an effective barrier to storm conditions, but the level of protection depends on geomorphic, hydrodynamic and ecological context of the location (Pinsky, Guannel & Arkema, 2013).

In addition, the coastal vulnerability mapping and modeling in this study was used to identify potential locations for habitat management or restoration to reduce coastal hazard risk to people and property. The input data and model limitations must be carefully considered when evaluating county-level decision-making about where to prioritize coastal management and restoration efforts. The coastal vulnerability assessment used the best available spatial data combined with local stakeholder input on several model parameters, but a number of assumptions and model limitations remain. This modeling effort allowed for an initial evaluation of the Marin county coastline in order to highlight locations and priority areas for more detailed and site-specific coastal modeling efforts necessary to provide more robust estimates of coastal erosion and inundation and improve future coastal vulnerability modeling efforts. Further, the InVEST approach is not a replacement for site-level hydrological analyses of inundation extent (e.g., Coastal Storm Modeling System) or habitat shifts (e.g., Sea Level Rise Affecting Marshes Model), as it provides a means for quickly comparing relative risk across a coastline and prioritizing areas for more detailed (and often more time-intensive) flood analysis. The map products created from the InVEST tool can provide the initial information necessary across a broad geographic extent to support the initial spatial evaluation of climate adaptation planning alternatives. Outputs can be used to better understand the relative contributions of these different model variables to coastal exposure and highlight the protective services offered by natural habitats to coastal populations. In particular, the model results may highlight change in coastal exposure with loss of habitat area. By coupling the exposure assessment mapping with land use planning spatial layers and framing the terminology to reference terms relevant to community planning, this information can help coastal managers, planners, landowners and other stakeholders begin to identify regions of relative greater risk to coastal hazards. This information can, in turn, better inform coastal resource use like development strategies and permitting.

We also evaluated the extent of these coastal protection benefits by mapping where the resulting estimates of high hazard exposure aligned with various land use zoning designations. Thus, we identified areas where people and property were exposed to coastal hazards in Marin county. This modeling and mapping approach allowed visual representation of the role that natural habitats play in reducing coastal exposure in Marin county and helped to inform priority locations for nature-based adaptation strategies during collaborative work with local planning agencies. InVEST is most effectively used within a decision-making process that starts with stakeholder consultations (Arkema et al., 2017; Arkema & Ruckelshaus, 2017). While nature-based strategies have gained in popularity, questions remain about how to best implement them as a component of coastal adaptation decision-making (Arkema & Ruckelshaus, 2017). Uncertainty persists regarding the effectiveness of the protective service of certain nature-based approaches when compared directly to an armored coastline, particularly when considering the spatial heterogeneity in the magnitude of protection provided (Koch et al., 2009). Though California understands the harm that hard armoring can inflict on adjacent ecosystems and public access points, and has even cautioned against using hard armoring altogether, the rate of armoring continues to increase along the coast due to the inequitable distribution of wealth, desire to delay inevitable retreat, and significant judicially imposed limits on the state’s ability to prevent coastal residents from armoring their property. The collaborative applied research approach adopted in Marin county showcases how to integrate nature-based solutions in the face of community conditions otherwise prone to hard armoring because of the nuanced, tailored research and associated results.

In collaboration with local decision makers, we linked our ecosystem service mapping and assessment to coastal adaptation decision making, and synthesized potential nature-based strategies relevant to these local coastal communities. By tailoring our mapping to the local area, we suggested management interventions which may have the highest likelihood of success in protecting people and the environment in this locality. Marin county is setting a precedent in updating their planning documents for climate adaptation in a way that takes ecosystem services analysis into account. This information can serve as a basis to determine where natural protections can be prioritized with inclusion of additional, localized considerations. The vulnerabilities identified in this process informed incorporation of appropriate coastal adaptation strategies and resilience measures into Marin county’s LCP amendments and update process as well as the county’s overall adaptation planning process—as specifically referenced in the “Adaptation Framework” section of the Marin Ocean Coast Sea Level Rise Adaptation Report (Marin County Community Development Agency, 2018). In addition, based on the findings from this report, the California State Coastal Conservancy awarded a grant for a Stinson Beach Nature-Based Adaptation Feasibility Study (Marin County Community Development Agency, 2021). In turn, the findings from that grant led to an additional “Coastal Resilience” grant from the California Ocean Protection Council to develop a long-term, implementable adaptation plan that includes the protective services provided by natural resources—ultimately informing an update to the county’s LCP.

Engagement between the planning community and members of the public is a pillar of the Local Coastal Program planning process. Members from Marin county’s planning offices spearheaded a significant public engagement effort on a range of topics, including the benefits of natural adaptation options. The figures and analysis from the coastal vulnerability modeling informed the production of materials and messaging points for public meetings and key stakeholder discussions. Furthermore, members of the planning community developed an engagement tool named “Game of Floods” to further educate audiences about the tradeoffs from pursuing specific adaptation activities—including those based on services from natural systems. These engagement tools and approaches helped facilitate dialogue between researchers, planners, and members of the local communities, ultimately leading to a more salient coastal analysis and planning process.

California has been engaged in adaptation planning for over fifty years (California Coastal Commission, 2018a). California’s Constitution and strong Public Trust Doctrine immortalize Californians’ right to public coastal access (Herzong & Hecht, 2013), and the California Coastal Act of 1976 creates a framework by which coastal municipalities must plan to adapt to climate change and manage coastal development. (California Public Resources Code, 1976). While there are policies in place at the state-level are intended to encourage prospective planning, a “one-size fits all model” of coastal adaptation is insufficient along the California coastline (Reiblich, Wedding & Hartge, 2017) because coastal jurisdictions vary in geomorphic characteristics (e.g., beaches, bluffs, estuaries), coastal and nearshore processes (e.g., waves, currents, sediment budgets), rates of sea-level rise (Griggs, 2017), as well as other factors, including unique cultures and political views. By co-developing our methodology with local planners in Marin county, against the backdrop of state-level guidance and community engagement, we tailored information and refined our analysis according to Marin county’s jurisdictional context and specific requirements—encompassing both rural and urban coastal communities with varied coastal landforms and ecosystems. The vulnerability modeling and policy analysis conducted in Marin county established methods and transferable approaches for incorporating coastal climate information into adaptation planning processes. By strategically considering multiple services provided by habitats when determining adaptation strategies, jurisdictions can work to protect people and property while also protecting or restoring dwindling critical habitat and the full suite of benefits those habitats provide to people (Heady et al., 2018; Sutton-Grier et al., 2018).

Conclusions

In this study, we linked our quantification of coastal ecosystem services directly to climate adaptation decision making, and highlighted the opportunity for nature-based strategies in two case study locations in California, USA. As a result of this information, policy recommendations included beach nourishment and dune restoration projects for the locations with dune habitats, and a horizontal levee for the wetlands. We anticipate that this approach will serve as a starting point and framework for further interdisciplinary work focused on bridging the gap between the best available science, law and policy in an iterative climate adaptation planning process at local scales. The adaptive capacity and sensitivity of coastal ecosystems vary greatly, and are also affected by management interventions (Morris et al., 2018) and climate change in combination with other anthropogenic stressors (Seddon et al., 2020).

Beyond California, climate change and adaptation planning are globally relevant issues in which the scalability and transferability of solutions should be considered. In the U.S., low-income and ethnic minority groups are disproportionately vulnerable to the effects of climate-induced coastal flooding, and this trend extends globally, where marginalized communities are often the most predisposed to climatic hazards (Reid et al., 2009). Future work could be expanded to recognize the importance and pervasive nature of environmental injustice in terms of coastal flooding. For instance, by coupling expanded model results with global population information, the model could show areas along a given coastline where humans are most vulnerable to storm waves and surge under different scenarios. This index has been used to evaluate the relative risk these hazards pose to different social groups as well as property (Arkema et al., 2013; Langridge et al., 2014; Ruckelshaus et al., 2016). Through this environmental justice lens, it is important to recognize that lower-income and ethnic minority coastal communities are disproportionately threatened by sea-level rise and coastal storms (Felsenstein & Lichter, 2014; Stallworthy, 2006). For instance, >99% of socially vulnerable people in Gulf regions of the U.S. live in areas which will likely not receive protection from coastal flooding (Martinich et al., 2013), and have already experienced the impact of Hurricane Katrina in 2005 and Hurricane Sandy in 2012.

Without climate adaptation efforts that incorporate conservation or restoration of coastal habitats, these ecosystems will continue to be lost, and their protective benefits will disappear with them. This work has outlined a framework for adaptation planning at a local-scale, and the next steps of this work could address the scalability of this iterative science-policy approach at the regional, national, and international scale. Over time, coastal communities like Marin will have “lessons learned” from the implementation of their nature-based adaptation planning. These lessons learned will then inform the next iteration of sea-level rise adaptation planning. Moving beyond the suitability and feasibility analysis of nature-based strategies (Reiblich et al., 2019), communities like Marin will soon be able to determine whether these strategies actually produced the intended results. On the leading edge of adaptation planning, many California coastal communities like Marin—rural and urban—find themselves pioneers in implementing sea-level rise adaptation strategies that incorporate nature-based strategies. Nature-based solutions such as these align with both national and sub-national long-term climate and biodiversity targets such as the USA’s Paris Agreement commitments, the 30x30 initiative, and the California Air Resources Board’s AB32 Climate Scoping Plan. Researchers and policymakers are urged to consider adaptation and mitigation strategies through nature-based infrastructure, which will be crucial in managing the impacts of climate change now and in the coming years as we tackle the climate and biodiversity crises (Seddon et al., 2020). The science-to-policy strategies outlined here can be a mechanism to engage community members, stakeholders, and decisionmakers through iterative, collaborative analyses and communication practices—ensuring an efficacious approach to address the full scope of the issue.

Supplemental Information

Supplemental Information 1 Supplemental Material

Click here for additional data file.

The authors would like to thank Jack Liebster and Alex Westhoff for their leadership and community engagement regarding coastal interests in Marin county, California. In addition, we thank Ross Clark for his passion and consultation on coastal adaptation efforts throughout the central coast of California. The authors also thank Sophie Taylor and Gillian Cowley for their background research in support of this work.

Additional Information and Declarations

Competing Interests

Author Contributions

Data Availability

The authors declare there are no competing interests.

Lisa M. Wedding conceived and designed the experiments, performed the experiments, analyzed the data, authored or reviewed drafts of the article, and approved the final draft.

Sarah Reiter conceived and designed the experiments, performed the experiments, analyzed the data, authored or reviewed drafts of the article, and approved the final draft.

Monica Moritsch performed the experiments, analyzed the data, prepared figures and/or tables, authored or reviewed drafts of the article, and approved the final draft.

Eric Hartge conceived and designed the experiments, authored or reviewed drafts of the article, and approved the final draft.

Jesse Reiblich performed the experiments, authored or reviewed drafts of the article, and approved the final draft.

Don Gourlie performed the experiments, authored or reviewed drafts of the article, and approved the final draft.

Anne Guerry conceived and designed the experiments, authored or reviewed drafts of the article, and approved the final draft.

The following information was supplied regarding data availability:

The data is available at figshare: Moritsch, M (2019): Marin_coastal_exposure.zip. figshare. Dataset. https://doi.org/10.6084/m9.figshare.7817981.v1.

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
