# Peer review of "Embedding the value of coastal ecosystem services into climate change adaptation planning"

_PeerJ, doi:10.7717/peerj.13463_

## Round 0.1 · original submission · Major Revisions

Dear Lisa and co-authors,

I have received three independent reviews of your study. While all reviewers have recognised the value of this work, they have also raised a number of issues that I consider major. In particular, all reviewers have noted that the manuscript should be considerably restructured to (1) provide clear research questions, (2) remove unnecessary discussions from the Methods section and provide a more focussed methodology in relation to your study, and (3) reduce the Discussion section and tie it more to the actual results produced in this study. They also noted that many typos occur throughout, and the revised version should be thoroughly proof-read.

Overall, reviewers have provided you with excellent suggestions on how to improve the next version. I will be looking forward to receiving your revised manuscript, along with a point-by-point response to the reviewers comments.

With warm regards,
Xavier

Reviewer 1 ·

Basic reporting

The paper does use professional English throughout, and the questions are clearly stated. However, I believe the article structure needs addressing. There is a lot of discussion within the methods and results which makes it difficult to discern what was actually done. It is unclear what raw data is used as this is mostly a qualitative analysis and there is no raw data provided.

The discussion is long and, particularly the policy discussion section, is rather ambiguous.

Experimental design

As is, I don't believe the manuscript fits within the Aims and Scope of PeerJ. Currently it is portrayed as a case study of two sites in California, and a lot of the focus is regarding the policy within these areas.

There is a lot of discussion within the methods, which makes it difficult to discern what was actually done. For example, L176-190 could be reduced and incorporated into the introduction, also the Policy context section. Much of the information in the Supplementary material is crucial for understanding how the model was run, and the manuscript would benefit from having most of those details included in the main methods, ie. the GIS Data inputs and and Exposure Index sections.

Validity of the findings

This is a qualitative study, so no underlying data has been provided and there are no definitive conclusions, rather discussion and recommendations.

As the aim of the manuscript is to link ecosystem services with policy about coastal adaptation, I believe the discussion could be improved by integrating the policy analysis discussion within the ecosystem service mapping and assessment. The policy discussion seems disconnected and is rather ambiguous, not really relating to the overall aim of the study. It would help to clarify the discussion by referring back to the original points stated in the last paragraph of the introduction.

Additional comments

This manuscript examines the coastal protection value of different habitats within two sites in California and displays the protective role of these habitats in terms of the land-use, using the InVEST model. It discusses this in the terms of current policy and gives recommendations on integrating such ecosystem measurements into policy. This topic is valuable and it's great to see a study trying to bridge policy and ecosystem services together. I do, however, feel that the clarity of the manuscript could be improved. It was difficult to work out what was done within the methods, and that made it hard to evaluate the results. The discussion is also quite long and would benefit from being sharpened and more targeted to the stated questions.

Major comments:
The Policy analysis is very qualitative and the section in the result appears to be more of a discussion with recommendations, rather than a statement of results. I think by integrating the results from the policy analysis with the results from the model, the value of the policy analysis would become clearer. At the moment it seems to be disconnected and much of the results of the policy analysis should be in the discussion rather than the results section.

The introduction is well written and clear. This paper would benefit from briefly introducing the InVEST model and the policy context within the introduction, but make sure to keep it brief to not swamp the introduction. Currently, the model and policies are discussed extensively within the materials & methods, which obscures what was actually done within the study.

In the abstract it is stated that the ecosystem services were quantified (Lines 58-58), however, as far as I can tell, this quantification is just the ranking on a scale of 1-5, which is already included within the model. I would argue that this is more of a qualitative or at most, a sub-quantitative assessment. At the end of the introduction it is stated that the natural habitats with protective benefits are identified and mapped, this is a more accurate way to describe what has been done.

Minor comments:
Line 73: sea grass -> seagrass

Fig. 1: If possible, it would make the figure clearer if the colour of the water was changed to grey so that the habitat types could stand out more.

Line 154: superscript 2 in km2

Line 168: Website address doesn’t work, needs to be: https://naturalcapitalproject.stanford.edu

Line 301: Two full stops used - “11.6% increase..”

Reviewer 2 ·

Basic reporting

Overall, there are a lot of typos throughout the manuscript and areas where the writing could be improved (see some specific suggestions below). This issue can be fixed by carefully reading and editing the manuscript before resubmission.

Experimental design

No comment.

Validity of the findings

No comment.

Additional comments

Summary:
This paper presents results from an analysis of the coastal protective benefits of natural habitats in Marin County, California using InVEST. This analysis is coupled with a policy analysis of different adaptation strategies that could be used in several case studies. Overall, I think this paper presents a valuable framework for bridging the gap between science and adaptation planning. I have only minor comments.

Minor comments:
Abstract: I think the abstract could use some work to improve clarity. See below.
Line 47: Change “the broad suite of benefits” to “a broad suite of benefits”
Line 48: Change “local-scale climate changed” to “local-scale climate change”
Line 53: Change “habitat” to “habitats”
Line 55: Is there supposed to be an “and” between erosion and inundation? If not, I’m not sure I know what erosion inundation is.
Line 55: Change “plays” to “play”
Lines 57-61: This may be a personal preference, but I think these lines should be present tense as they are not analyses that you did but rather discussions that the present ms is covering
Line 61: Change “We anticipate this approach as a starting point” to “We anticipate that this approach will serve as a starting point”
Line 67: Change “has outlined” to “outlines”
Line 68: Change “scalability to the regional, national, and international scale” to “scalability to regional, national, and international scales”

Introduction:
Line 73: Change “sea grass” to “seagrasses” for consistency
Line 75: Delete the word locations
Line 85: Change “these” to “natural”
Line 88: Change “Installation of hardened shoreline structures protects structures immediately behind them but alter sediment transport regimes, eventually leading to beach erosion in front of and downcoast of armoring” to “Hardened shoreline structures can protect infrastructure immediately behind them but can also alter sediment transport regimes, eventually leading to beach erosion in front of and adjacent to armoring”
Line 91: Change “maintain integrity” to “maintain the integrity”
Line 99: Change “land use and protection” to “land use via protection”
102: incorporating into what? This sentence is incomplete
Lines 105-108: This doesn’t seem like a great example, as it is just an example of restoration rather than nature-based infrastructure
Lines 114-115: Redundant. This is the 4th time just in the introduction that it has been mentioned that habitats provide recreation and wildlife habitat.
Lines 119-32: Excellent! Perfectly sums up the intention of the paper and the different components. I think the abstract needs to be re-written with this paragraph in mind in order to better emphasize the study objectives and different components of the study.

Methods:
Lines 140-143: Redundant, this could be combined into one sentence
Line 179: Change “strategy approaches” to “strategies”
Line 181: Change “versus nature” to “versus natural”
Line 274: Be consistent with whether or not you capitalize “county”

Results:
Line 294: Delete the period before the figure reference
Line 299: Delete “of”
Line 301: Delete duplicate period
Line 348: Dune is misspelled
Line 360: Change “planning In Dillon” to “planning. In Dillon”
Lines 384-385: Did this sentence get cut off?

Discussion:
Again, this may be preference, but I think that the results should be referred to in present tense, not past
Line 400: Change “the delivery of these, and other, critical ecosystem services” to “the delivery of critical ecosystem services”
Line 403: Change “can be the basis” to “can serve as a basis”
Line 4011: Change “habitat” to “habitats”
Lines 398-432: This section needs to be re-written with greater attention to interpreting the results from the paper rather than describing what the InVEST tool can do. How does this analysis compare with other studies looking at the hazard mitigation value of different coastal habitats?
Line 477: Decision-making needs a hyphen
Line 490: Change “implementation” to “implementing”
Line 493: Be consistent about whether or not you hyphenate SLR

Figures and Tables:
Table S2: Where are these estimates coming from? Seems like there should be some citations to go along with these. What does habitat rank mean? The risk of erosion?

·

Basic reporting

The writing is relatively clear, but there are some general statements that seem kind of meaningless and confusing, and a fair amount of repetitive introductory text in various sections.
The intro and background section describe the context and general literature well. They tend to be a bit general however, not covering other literature on ecosystem services and land use mapping. There is a significant literature on this from the UK and EU.
The structure appears to conform to standard.
The maps included are very nicely done, but it is not clear to me if figure 1 and figure 4 are needed. It seems that this analysis is really about the more specific areas showing in figures 2 and 3.
Data is provided.

Experimental design

Article is within scope of journal.
Research question is not clearly stated, rather they seem to be exploring the potential application of research in local decision making.
The method used for the coastal protection assessment was well applied. While not the most rigorous method available to assess how habitat contributes to coastal protection, it is a reasonable choice for a low capacity setting where the results may be “good enough” to inform planning. It would be useful for the authors to describe this more – that there are more rigorous engineering model-based methods but that this method was selected for specific reasons. They appear to be missing a methodology to assess the impact of their work on the planning process or outcomes.
Methods used were described clearly and in a replicable manner.

Validity of the findings

Underlying data were provided and modeling followed published methods. A limitation of these methods and the use of an index is that there is no clear articulation of uncertainty or how it may impact results.
The results section was a straightforward description of the coastal exposure mapping but did not clearly explain the integration with planning – either how planning influenced the mapping, or how the mapping influenced planning. Much of the discussion was broad and not well tied to research results.
Conclusions were very general and repeat general study assumptions, rather than specific results.

Additional comments

Many local coastal adaptation planning processes and tools do not consider the benefits provided by natural habitats, thus I appreciate the authors' intentions to help provide such information for local planning. I have four major suggestions.
1) I think the authors need to better articulate their research question and then provide results that directly correspond. For example, if the question is “Can mapping the coastal protection benefit of natural habitats result in a more intentional inclusion of natural habitats in local coastal adaptation planning?”, there need to be results about how the planning process was changed by the inclusion of the new maps.
2) While the results explained what planners had decided to do with natural habitats, there is no explanation of the role the new coastal protection maps played. It is important that the results include some indication of whether or how the availability of these maps changed the planning process or outcomes. Ideally this would include some formal methodology and process – perhaps structured interviews of those involved in the planning process and decisions.
3) The Authors also need to clarify how their coastal protection mapping methods were adapted to link to the local planning process.
4) The authors mention in a couple places that they ran their models under different sea level rise scenarios but they do not provide these results anywhere, or explain how this was used in the analysis or planning. Sea level rise is the primary issue adaptation planners need to consider. In many places the coast line will move and change so much that current habitat maps are not relevant to future habitats and the coastal protection they might provide. In many cases adaptation planning is focused more on identifying habitat migration corridors than on the benefits of current habitats. So, there needs to be much greater discussion of this issue and this part of the analysis.
Detailed comments were provided on the PDF manuscript.

---

## Round 0.2 · Minor Revisions

Dear Lisa and co-authors,

First, my apologies for the delay in response - One pending reviewer is not available anymore. Therefore, I am taking this decision based on a single reviewer who has provided you with some excellent additional comments that will further improve your revised manuscript. I recommend minor revisions, but I would actually say a moderate level of revisions are needed to make this paper as clear and effective as it could be, as indicated by the reviewer.

Please note that the reviewer has included additional comments via the attached annotated pdf.

I'll be looking forward to receiving your final revised manuscript, along with point-by-point responses to the reviewer's comments.

With warm regards,
Xavier

·

Basic reporting

Writing is relatively clear overall; see minor suggested improvements throughout paper.
Background information provided and literature referenced in the introduction is relevant and sets up the need for information about shoreline protection by coastal habitats. The context about climate adaptation and relevant policies in California is particularly helpful in explaining why LCPs are a good engagement point for this type of information – most of the details about this are in the methods section (starting with line 157) and may fit better in the introduction.
Maps are relevant and attractive.
Raw data is not presented in the supplementary materials but appear to be available somewhere. Maybe clarify in the paper.

Experimental design

Yes, this is original research within scope for the journal. The research questions are stated and put into context of coastal community partners in Marin county and may be a model for application in other coastal communities.
Generally the study is performed well but a few questions remain about the coastal vulnerability modeling – these may be cleared up with better description in the methods about how community engagement was used to inform modeling:
• It is not clear how protective ranks/distances of various habitat types were assigned in the InVEST model (see comment on lines 221-222)
• Dividing coastal areas into quantiles based on sea level rise projections (for the SLR factor in the model) may not be appropriate depending on the range of projected SLR elevations in the study area – this could affect the observed geographic differences in the exposure index (see comment on lines 228-229)

Methods describing individual aspects of this project (the InVEST modeling, policy analysis) are generally sufficient, but connections between these aspects are missing. For example, the methods do not describe how coastal vulnerability mapping was used to identify locations or options for habitat management/restoration to reduce risk. The methods also do not describe engagement with Marin county planners or community, although the discussion states that engagement guided method development and helped to integrate results into the planning process (lines 383-386, 409-411).

Validity of the findings

• The type of information/conclusions that can be made from the InVEST coastal vulnerability model in general, and the modeling done in this project specifically, is sometimes stated in ways that will cause readers to draw inaccurate conclusions (see comments on lines 115-116, 345-348, 351)
• Results section could better differentiate between the coastal vulnerability mapping results and the policy analysis results. Currently, the two blend together starting around line 301.
• It is not clear whether/how the results of the coastal vulnerability mapping were used to identify potential policy/planning options.

Additional comments

• Incorporating the function of coastal habitats for protection of coastal infrastructure and communities into strategic planning is valuable, and using a mapping approach to consider options in a particular context is an excellent way to go about that. However, as written this paper does not fully explain how this was done in Marin county, and therefore is less useful as an example for other researchers or communities hoping to use a similar approach. Better explanation of what engagement with planners or the community took place, and of how the mapping was used to inform the policy/management aspects of this work would be a big improvement.
• Are the rankings for the relief factor reversed in Table S1? Low-relief areas (0th-20th percentile) should have the highest risk of inundation rather than the lowest.

---

## Round 0.3 · accepted · Accept

Dear Lisa and co-authors,

I have now carefully reviewed your rebuttal and revised (v2) manuscript, and am please to accept it for publication in PeerJ. I have attached an editor annotated version of the manuscript, including small suggested corrections that you may address at the proof stage. There are lots of formatting issues with your references, but the PeerJ staff will contact you with specific questions.

I would like to thank the reviewers for their time and efforts in improving the manuscript to this stage, and thank you authors for this great contribution to the field!

With warm regards,
Xavier